# Genomic regions controlling shape variation in the first upper molar of the house mouse

**Luisa F Pallares[1†‡]\*, Ronan Ledevin[2†§], Sophie Pantalacci[3], Leslie M Turner[1,4], Eirikur Steingrimsson[5], Sabrina Renaud[2]**

[1]Department of Evolutionary Genetics, Max-Planck Institute for Evolutionary Biology, Plön, Germany; [2]Laboratoire de Biométrie et Biologie Evolutive, UMR5558, CNRS, University Lyon 1, Campus de la Doua, Villeurbanne, France; [3]ENS de Lyon, Univ Claude Bernard, CNRS UMR 5239, INSERM U1210, Laboratoire de Biologie et Modélisation de la Cellule, 15 parvis Descartes, F-69007, UnivLyon, Lyon, France; [4]Department of Biology and Biochemistry, Milner Centre for Evolution, University of Bath, Bath, Unites States; [5]Department of Biochemistry and Molecular Biology, BioMedical Center, Faculty of Medicine, University of Iceland, Reykjavik, Iceland

**\*For correspondence:**
pallares@princeton.edu

[†]These authors contributed equally to this work

**Present address:** [‡]Lewis-Sigler Institute for Integrative Genomics, Princeton University, Princeton, United States; [§]Université de Bordeaux, PACEA, UMR5199, 33615 Pessac, France

**Competing interests:** The authors declare that no competing interests exist.

**Abstract** Numerous loci of large effect have been shown to underlie phenotypic variation between species. However, loci with subtle effects are presumably more frequently involved in microevolutionary processes but have rarely been discovered. We explore the genetic basis of shape variation in the first upper molar of hybrid mice between *Mus musculus musculus* and *M. m. domesticus*. We performed the first genome-wide association study for molar shape and used 3D surface morphometrics to quantify subtle variation between individuals. We show that many loci of small effect underlie phenotypic variation, and identify five genomic regions associated with tooth shape; one region contained the gene microphthalmia-associated transcription factor *Mitf* that has previously been associated with tooth malformations. Using a panel of five mutant laboratory strains, we show the effect of the *Mitf* gene on tooth shape. This is the first report of a gene causing subtle but consistent variation in tooth shape resembling variation in nature.
DOI: https://doi.org/10.7554/eLife.29510.001

## Introduction

Understanding the genetic basis of evolution requires the identification of genes and mutations responsible for phenotypic variation between individuals and populations. A complete catalog of such variants will allow the identification of the genetic paths favored by evolution (*Stern and Orgogozo, 2008*; *Martin and Orgogozo, 2013*). Regarding morphological variation, the latest catalog, that includes animals, yeasts, and plants, listed 386 alleles (*Martin and Orgogozo, 2013*), most of them related to pigmentation. Genes associated with shape variation are fewer, and correspond to variation in wing shape in flies and butterflies, and body shape in fish. However, there have been many other studies exploring the genetic basis of shape variation (e.g. craniofacial shape), but given the highly polygenic nature of these traits it has been difficult to validate the extensive list of candidate genes identified through quantitative trait loci (QTL) and genome-wide association (GWAS) approaches.

The mouse tooth is one of the traits extensively studied regarding shape variation. In paleontology, it is a key character for phylogenetic and dietary inferences (*Misonne, 1969*; *Gómez Cano et al., 2013*). It has been a model for developmental genetics (*Jernvall and Thesleff, 2012*; *Urdy et al., 2016*). The genes and pathways involved in tooth morphogenesis are well known

(*Thesleff and Sharpe, 1997*; *Jernvall and Thesleff, 2000*; *Jernvall and Thesleff, 2012*; *Lan et al., 2014*) and a few genes affecting cusp patterning in mice have been identified (*Jernvall and Thesleff, 2012*). Moreover, computational models have been able to recreate the morphological transition between species and individuals (*Salazar-Ciudad and Jernvall, 2002, 2010*; *Harjunmaa et al., 2014*; *Urdy et al., 2016*). As for many other traits, there is a wide gap between the macro- and micro-evolutionary understanding of tooth morphological variation (*Nunes et al., 2013*). The widespread morphological variation at the micro-evolutionary level in mice (i.e. within-species) has been repeatedly highlighted (*Boell and Tautz, 2011*; *Ledevin et al., 2016*), as well as its relevance for understanding the evolutionary history of the taxon (*Macholán, 2006*; *Renaud et al., 2011*; *Renaud and Auffray, 2013*; *Renaud et al., 2013*).

Surprisingly, only two studies have tried to map loci responsible for such morphological variation. Using F2 crosses between LG/J and SM/J inbred lines of mice, *Workman et al., 2002* found three QTL for molar row size, and 18 QTL for 2D molar row shape. Using recombinant inbred lines between A/J and SM/J, *Shimizu et al. (2004)* found seven QTL affecting size of either the upper or lower molars. As is common for QTL approaches, hundreds of genes co-localized with the QTL, impeding the identification of clear candidate loci.

In this study, we focused on the first upper molar and used mice derived from a wild population, which allowed us to directly address the genetic basis of molar shape variation from a micro-evolutionary perspective. The mice used here are first-generation offspring of natural hybrids between *Mus musculus musculus* and *Mus musculus domesticus* collected in the Bavarian hybrid zone (*Turner et al., 2012*). As a result, phenotypic variation in the sample has a between-subspecies as well as a within-population component. This sample represents phenotypic and genetic variation segregating in nature, and therefore constitutes an evolutionarily relevant scenario in contrast to inbred lines traditionally used in mapping studies in mice. Previous studies successfully identified loci associated with cranial morphology and male sterility phenotypes in this mapping population (*Pallares et al., 2014*; *Turner and Harr, 2014*). We performed the first GWAS of 3D molar shape variation, and identified several candidate regions for naturally occurring variation. We also, for the first time, showed that mutations in the candidate gene *Mitf* can cause between-individual differences in molar shape in the mouse.

## Materials and methods

### Samples used in the mapping

The mice used in this study are laboratory-bred first-generation offspring of matings between wild hybrid mice caught in the Bavarian hybrid zone between *Mus musculus musculus* and *Mus musculus domesticus* (*Turner et al., 2012*). See *Turner et al. (2012)* for details on animal experiments and ethics. Further details were provided in previous mapping studies of skull and mandible shape (*Pallares et al., 2014*), and male sterility phenotypes (*Turner and Harr, 2014*). All mice were raised under controlled laboratory conditions and are males between 9 and 12 weeks old; the sample used here includes siblings and half-siblings.

### 3D surface morphometrics

Mice heads were scanned at a cubic resolution of 0.021 mm using a vivaCT 40 micro-computer tomograph (Scanco, Bruettisellen, Switzerland) (*Pallares et al., 2014*). Three-dimensional (3D) virtual surfaces of the left first upper molar (UM1) were generated for all specimens using a semi-automatic segmentation in Avizo software (v8.1 - Visualization Sciences Group, FEI Company). The connections between the tooth and the surrounding materials (i.e. second upper molar and maxillary bone) were manually closed.

We focused on the first upper molar given that it is the first molar to develop, and therefore it is not directly constrained by the development of other molars, as is the case for the second and third molars (*Kavanagh et al., 2007*). In addition, upper molars are morphologically more complex than lower molars. We expect that those two factors are reflected in a stronger genetic signal and higher phenotypic variance, respectively, relative to other molar teeth.

To quantify the 3D shape of the molar tooth, a template was designed based on a randomly chosen tooth. The template corresponds to the surface describing the erupted part of the tooth, with

the roots and UM1/UM2 junction manually removed. On this surface, 1500 equally spaced 3D semi-landmarks were requested, resulting in 1588 automatically sampled points. Ten landmarks were placed manually on the tooth surface using the Landmark Editor software (version 3.0.0.2, Institute for Data Analysis and Visualization) and were used to anchor the template on the original surface. For each specimen, the template was then deformed to match the tooth surface using the R package Morpho (*Schlager, 2016*).

The semi-landmarks were allowed to slide along tangent planes to the surface using the minimum bending energy criterion, that is minimizing the bending energy necessary to produce the changes in each specimen relative to the Procrustes consensus configuration (*Bookstein, 1997*; *Gunz and Mitteroecker, 2005*). Specimens were then aligned using a full Procrustes superimposition. In this way, all differences due to scale, position, and orientation were removed, and the shape variables (Procrustes coordinates) describing tooth shape were extracted.

In addition, we used an approach that directly addresses the effect of wear on tooth shape. It is known that the degree of wear is a major cause of non-heritable shape variation. Following *Ledevin et al. (2016)*, a template was designed with the top of the cusps cut to mimic a fixed degree of wear. The height at which the template was cut was decided empirically, assuring that the more worn teeth in our dataset were still able to be described with the template. We will refer to it as 'wear-free template'. On this new template, again, 1500 equally spaced semi-landmarks were requested, resulting in 1532 points being automatically sampled and anchored by seven landmarks. The same procedure described above was used to obtain shape variables.

For each data set (complete and wear-free template), a principal component analysis (PCA) was performed on the covariance matrix of shape variables. Because of the large number of variables (more than 1000 points in three dimensions), a reduction of dimensionality (e.g. Sheets et al. 2006) was performed. We used PCs representing >1% of variance as phenotypes for mapping; this dataset comprises 18 PCs representing 81.4% and 86.6% of total shape variation in the complete and wear-free template, respectively.

The relationship between shape and age was investigated using a multivariate regression of the 18 PCs on mouse age.

Centroid size (CS), estimated as the square root of the summed squared distance between each semi-landmark and the centroid, was used as indicator of tooth size.

## Genotypes

SNP genotypes for the 183 mice used in the mapping were obtained from *Turner and Harr, 2014a*. Details on SNP quality control can be found in *Turner and Harr, 2014a*. In short, 584 729 SNPs were genotyped using the Mouse Diversity Genotyping Array (Affymetrix, Santa Clara, CA) (*Yang et al., 2009*). SNPs with heterozygosity >0.9, with >5% missing data, or minor allele frequency <5% were removed. SNPs in perfect linkage disequilibrium (LD) with other SNPs were filtered, resulting in 145 378 SNPs used for association mapping. The full set of SNPs previous to LD pruning was obtained from *Pallares and Harr, 2014*.

## Association mapping

183 mice, 145 378 SNPs, and 18 PCs, each representing more than 1% of shape variation were used to map loci associated with shape variation of the first upper molar. Each PC was analyzed separately, and therefore, the approach used here corresponds to a univariate mapping of shape variables (PCs). This approach was used in order to implement a linear mixed-model (LMM) to control for family structure (see below); however, it should be noted that mapping PCs might not identify genetic associations that are not strongly aligned with single PCs but that are spread across multiple PCs. CS was used to find associations with size variation. The PC scores and CS data used for the mapping are available as *Figure 4—source data 1*.

We performed mapping using the LMM implemented in GEMMA (*Zhou and Stephens, 2012*). A centered kinship matrix was used to correct for relatedness between individuals and hidden population structure. A genome-wide significance threshold was calculated for each phenotype (PCs and CS) using permutations as explained in *Pallares et al., 2014*. In short, a distribution of the best p-values was generated based on 10,000 permutations of the phenotypes, the 95% of such

distribution was used as significance threshold. The permutations for the chromosome X were performed separately from the autosomes (*Turner and Harr, 2014*).

The linkage disequilibrium (LD) between the significant SNPs and neighboring SNPs (full dataset, prior to LD pruning) was used to delimit the significant regions associated with molar shape variation. A threshold of $r^2 > 0.8$ was used. For significant SNPs without neighboring SNPs in tight linkage, we report regions extending 250 Kb to each side of the best (lowest p-value) SNP resulting in regions of 500 Kb. This value is based on results from a previous study mapping cranial morphology in this sample (*Pallares et al., 2014*); intermediate between the median size of regions tightly linked ($r^2 > 0.8$, 150 kb) and weakly linked ($r^2 > 0.2$, 1.8 Mb) to significant SNPs. LD calculations were done in PLINK 1.07 (*Purcell et al., 2007*).

The effect of the significant QTLs on the phenotype was calculated based on Procrustes distances as the coefficient of determination ($r^2$) between the 18 PCs and the genotype of the best SNP per region. In this way, the effect size of an SNP is relative to total phenotypic variation, and not to the individual PC it was associated with in the mapping. It should be kept in mind that given that siblings and half-siblings were used for mapping, the coefficient of determination could result in an overestimation of the effect size.

## SNP heritability

SNP heritability is the amount of phenotypic variation explained by the additive effect of all SNPs used in the mapping (*Wray et al., 2013*). This value is a proxy for the amount of additive genetic variance in the sample. We estimated SNP heritability for each PC under the linear mixed-model (LMM) in GEMMA ('pve' - percentage of variance explained) (*Zhou and Stephens, 2012*); the weighted sum of the PCs heritability was used as a proxy for the total SNP heritability of molar shape in this population of mice. The weight was given by the percentage of phenotypic variation represented by each PC. Here, we have opted for estimating heritability as a scalar value (*Monteiro et al., 2002*; *Monteiro et al., 2003*). In this way, we are able to estimate the contribution of genetic variance to overall shape variation in this sample, and to make this value comparable to other studies.

To estimate the proportion of phenotypic variation explained by each chromosome, we used the restricted maximum-likelihood (REML) analysis implemented in GCTA (*Yang et al., 2011*). Due to the small sample size in this study, each chromosome was analyzed separately including the first 10 PCs of the kinship matrix as covariates (option–reml–grm–qcovar). As a result of not fitting all chromosomes at the same time, values for individual chromosomes were inflated resulting in a larger SNP heritability than the one calculated with all SNPs at the same time (see above). Therefore, throughout the manuscript, we used the relative contribution of each chromosome to the phenotype, instead of the absolute value.

## Samples used to functionally evaluate Mitf

To further explore the role of the candidate gene *microphthalmia* (*Mitf*) in molar shape variation, we took advantage of four existing laboratory mouse strains carrying mutant alleles, including *Mitf^mi-vga9*, *Mitf^mi-enu22(398)*, *Mitf^Mi-wh*, and *Mitf^mi*. Details about each mutant allele and associated phenotypes are reported in *Table 1*. All mice were raised at the University of Iceland, BioMedical Center, under permit number 2013-03-01 from the Committee on Experimental Animals (Tilraunadýranefnd).

Homozygous, heterozygous, and wild-type mice were collected to test for differences in molar shape. The total sample size consisted of 36 mice: five *Mitf^mi-vga9*/+, five *Mitf^mi-vga9*/*Mitf^mi-vga9*, 10 *Mitf^mi-enu22(398)*/*Mitf^mi-enu22(398)*, two *Mitf^Mi-wh*/+, four *Mitf^Mi-wh*/*Mitf^Mi-wh*, and five compound heterozygotes *Mitf^Mi-wh*/*Mitf^mi*. All these mutations are on C57Bl/6J (B6) background, and therefore five B6 mice were used as the wild-type control. Heterozygous and homozygous mice for the *Mitf^Mi-wh* allele were siblings, as well as mice with the *Mitf^mi-vga9* allele. The mutant mice were male and female ranging from 5 to 10 weeks. Due to the small sample size per group, it was not possible to test for sexual dimorphism in molar shape; however, it has been shown that sex has very small effect, if at all, on tooth shape in mice (e.g. *Valenzuela-Lamas et al., 2011*; *Renaud et al., 2017*). To control for age and related wear effects, the heads were scanned and phenotyped as described above using the wear-free template.

**Table 1.** *Mitf* alleles used in this study.

The effect on gene expression as well as the organismal phenotype associated with each allele is shown. All mutants are on C57Bl/6J background.

| Allele | Symbol | Mode of induction | Lesion | Effect | Phenotype Heterozygote | Homozygote |
|---|---|---|---|---|---|---|
| micropthalmia | *Mitf*$^{mi}$ | X-irradiation | 3 bp deletion in basic domain | Affects *Mitf* DNA binding affinity | Iris pigment less than in wild type; spots on belly, head and tail | White coat, eyes small and red; deficiency of mast cells, basophils, and natural killer cells; spinal ganglia, adrenal medulla, and dermis smaller than normal; incisors fail to erupt, osteopetrosis; inner ear defects |
| White | *Mitf*$^{Mi-wh}$ | Spontaneous or X-irradiation | I212N | Affects Mitf DNA binding affinity | Coat color lighter than dilute (*d/d*); eyes dark ruby; spots on feet, tail and belly; inner ear defects | White coat; eyes small and slightly pigmented; spinal ganglia, adrenal medulla, and dermis smaller than normal; inner ear defects; reduced fertility |
| VGA-9 | *Mitf*$^{mi-vga9}$ | Transgene insertion | Transgene insertion and 882 bp deletion | Loss-of-function | Normal | White coat, eyes red and small; inner ear defects |
| enu-22(398) | *Mitf*$^{mi-enu22(398)}$ | Ethylnitroso-urea | C205T, Q26STOP in exon 2A, unpigmented spots in coat | Affects splicing | Normal | Normal eyes, white belly and large |

This table was modified from *Steingrimsson et al. (2004)*. Information for the allele enu-22(398) comes from *Bauer et al. (2009)*.

DOI: https://doi.org/10.7554/eLife.29510.002

## Functional evaluation of the candidate gene Mitf

A PCA including all *Mitf* mutant and wild-type mice was performed to explore and visualize shape variation. However, to test for significant effects of each mutation on molar shape, a PCA was performed with pair of groups involving a mutant genotype at a time and the wild-type B6 mice. Heterozygous and homozygous mice for the same allele were tested independently against wild-type B6 group. The first two PCs of each PCA were used in a Hotelling $T^2$-test to assess the significance of mean shape differences between the mutant groups and WT mice (R function *hotelling.test*). Since there are only two *Mitf*$^{Mi-wh}$/+ mice, they were not included in this analysis. p-Values were corrected for multiple testing using the Holm-Bonferroni method in R.

The comparison between the morphological changes associated with each mutant and the effects associated with the SNP identified in the mapping encountered the problem that each analysis was done with independent Procrustes superimpositions. This means that the PC axes from both analyses are not comparable. We therefore developed an approach to be able to compare the morphological signature of each effect on the tooth.

Shape changes were visualized using reconstructed surfaces (e.g. for a group consensus, or along PCs). Then, for two surfaces, the distance between each vertex was calculated. The correlation between the effect of an SNP and (i) a PC or (ii) a mutation was assessed as follows: (1) Each effect was characterized by a pair of reconstructed surfaces: a surface for each homozygous genotype; the mean surface of the wild-type vs the mean surface of a mutant; the shapes reconstructed from the scores at the two extremes of each PC used in the mapping. (2) For each pair of surfaces, the distances for each of their vertices were calculated. This provided, for each effect, a range of 8150 between-vertices distances. (3) The between-vertices distances were compared between two effects: if they are similar, a large difference in a part of the tooth for one effect should also correspond to a large difference for the other effect. This provided a quantitative indicator of the degree of resemblance between two factors (e.g. an SNP and mutation or a PC). (4) To generate a proxy for the correlation to be expected between orthogonal directions of change, the shape changes represented by the PCs used in the mapping were compared to each other. However, it should be kept in mind that although PC axes are statistically independent, this is not necessarily true regarding the underlying genetics.

## Results

The mice used in this study were derived from wild-caught hybrid mice between *M. m. musculus* and *M. m. domesticus* and represent a hybridization continuum between the two subspecies (*Figure 1*, a). Molars of *M. m. musculus* are characterized by anterior elongation, expansion of the labial cusps, and reduction of the antero-lingual cusp compared to *M. m. domesticus* mice (*Figure 1*, b). Despite the hybrid character of the sample, the major axes of variation are not polarized by shape differences between the two subspecies (*Figure 1*, c). This indicates that phenotypic variation in the sample has a between-subspecies component, but axes of within-subspecies variation are more important, representing other directions of shape changes different from *M. m. musculus* – *M. m. domesticus*. This suggests that within-population variation in molar shape is larger than between-subspecies variation, and therefore species-specific alleles might not be playing any major role in tooth shape in this population. A similar pattern of between- vs within-population variation has been previously described for mandible shape (*Boell and Tautz, 2011*). Additional factors that might contribute to such a pattern are transgressive phenotypes and hybrid developmental instability, although the latter seems to be not very important in this population (see *Pallares et al., 2016*).

### Tooth shape, mouse age and wear

Once erupted, tooth shape does not change except by the effect of wear, but the degree of wear is correlated with the age of the mice, that in this study ranges from 9 to 12 weeks. The effect of age and wear on molar tooth shape was explored using the first 18 PCs derived from each of the two approaches: complete template and wear-free template. Age differences have a small but significant effect on molar shape variation when using the complete template (p-value=$5.4\times10^{-5}$, $r^2$=0.01), and this is reflected in the significant correlation between age and some PCs ($r^2$(PC1) = 2.1%, p-value=0.027; $r^2$(PC4) = 2.2%, p-value=0.025; $r^2$(PC5) = 3.3%, p-value=0.008; $r^2$(PC14) = 1.8%, p-value=0.04; $r^2$(PC18) = 4.9%, p-value=0.002). More importantly, a qualitative assessment indicates that wear-like patterns are present already in the first axis of variation (PC1) (*Figure 2*-a).

Although age correlates with wear patterns, there are other factors, such as diet and behavior that also influence tooth morphology through wear (*Renaud and Ledevin, 2017*). The wear-free template lacks the tip of the cusps, mimicking the same level of wear in all individuals (*Figure 2*-b), and therefore addressing wear in all its complexity, being this related to age or any other factors. The effect of age on overall shape variation using this template was still significant but explained very little variance (p-value=0.01, $r^2$=0.01). Only two PCs kept a very small age-related signal ($r^2$(PC3) = 4.6%, p-value=0.002; $r^2$(PC18) = 3.5, p-value=0.007).

### Heritability

The heritability values estimated in this study correspond to the effect that all SNPs used in the mapping have on the phenotype; this value is also known as SNP or chip heritability, and it serves as a proxy of the additive genetic variance underlying phenotypic variation. Substantial genetic variance was found in all PCs (*Table 2*). A weighted average of all PCs genetic variance was used to

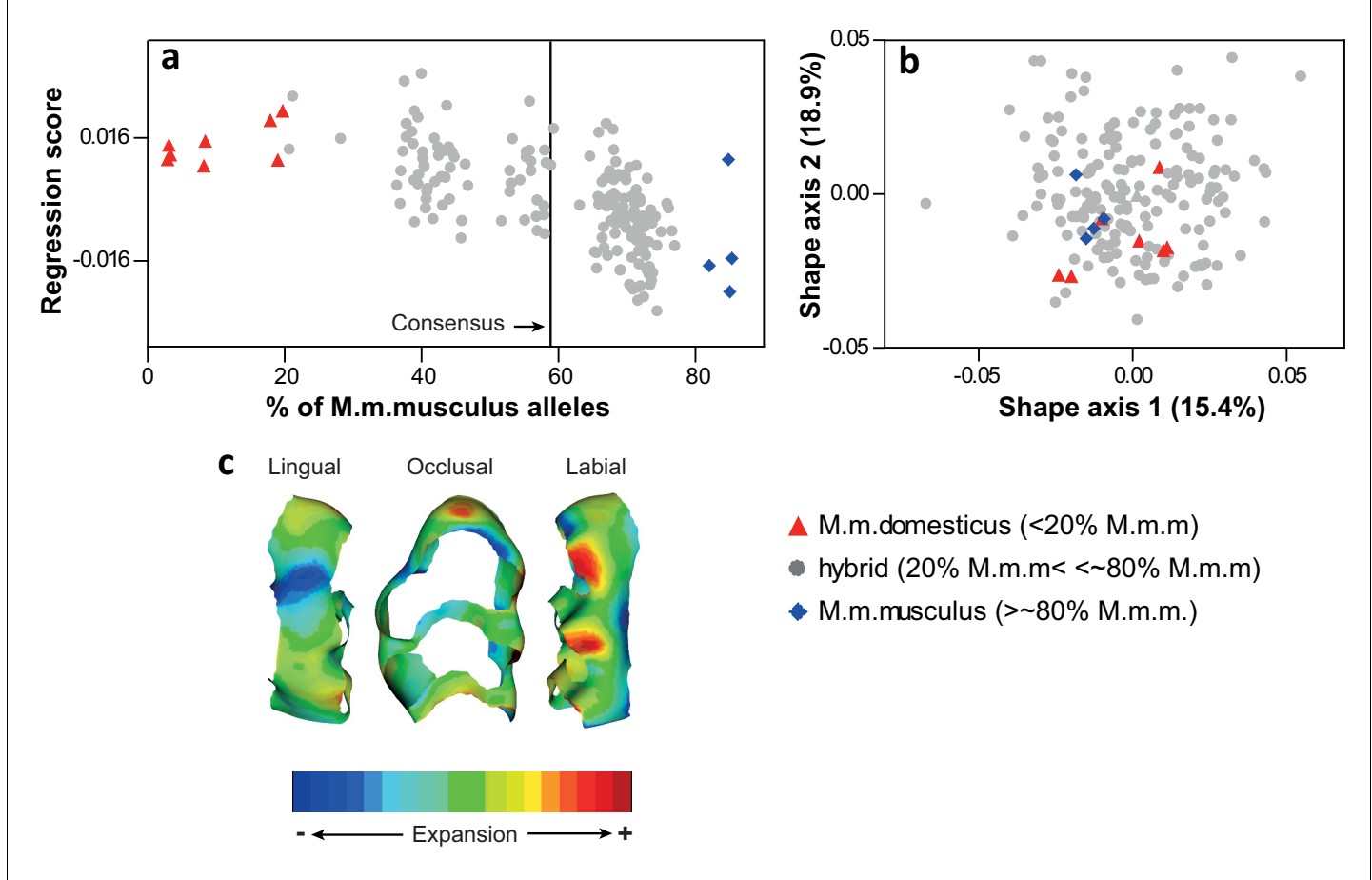

**Figure 1.** Molar shape variation in the sample. (**a**) Multivariate regression of molar shape on the degree of hybridization (*M.m.musculus* ancestry per individual was obtained from *Turner et al. (2012)*). (**b**) Shape variation in the sample depicted on the first two principal axes of a PCA. (**c**) Transition in molar shape from *M.m.domesticus* to *M.m.musculus.* All shape data were obtained from the wear-free template.
DOI: https://doi.org/10.7554/eLife.29510.003

summarize the total heritability of molar shape (see Materials and methods). This resulted in a value of 65.5%, indicating that more than half of shape variation is accounted for by additive genetic effects.

When exploring heritability from a chromosomal point of view (*Figure 3*), instead of individual associations between SNPs and phenotype, a positive correlation is evident between the amount of phenotypic variation explained by each chromosome and chromosomal length (r = 0.67, p-value=0.001). This is the expected pattern when the effect of individual SNPs is small, and such SNPs are many and homogeneously distributed along the genome.

## Mapping of loci associated with molar shape variation

We decomposed variation in molar tooth shape in PCs. The mapping was performed using 183 mice and ~145,000 SNPs. Centroid size was used as proxy for tooth size, however, no genomic regions were significantly associated with size variation.

When shape data derived from the complete template was used in the mapping, no significant associations were found. In contrast, the data derived from the wear-free template resulted in nine SNPs significantly associated with molar shape variation, clustered in five genomic regions. Hereafter, we will thus focus on the results of the wear-free approach.

These five loci are found in chromosomes 1, 5, 6, and X, and were associated with PC7, PC11, PC16, and PC18 (*Figure 4*-a, *Table 3*). The name of each locus is defined by Mo (molar) and chromosomal location. Regions Mo.1, Mo.5, Mo.6, and Mo.X.1 have an arbitrary size of 500 Kb (see

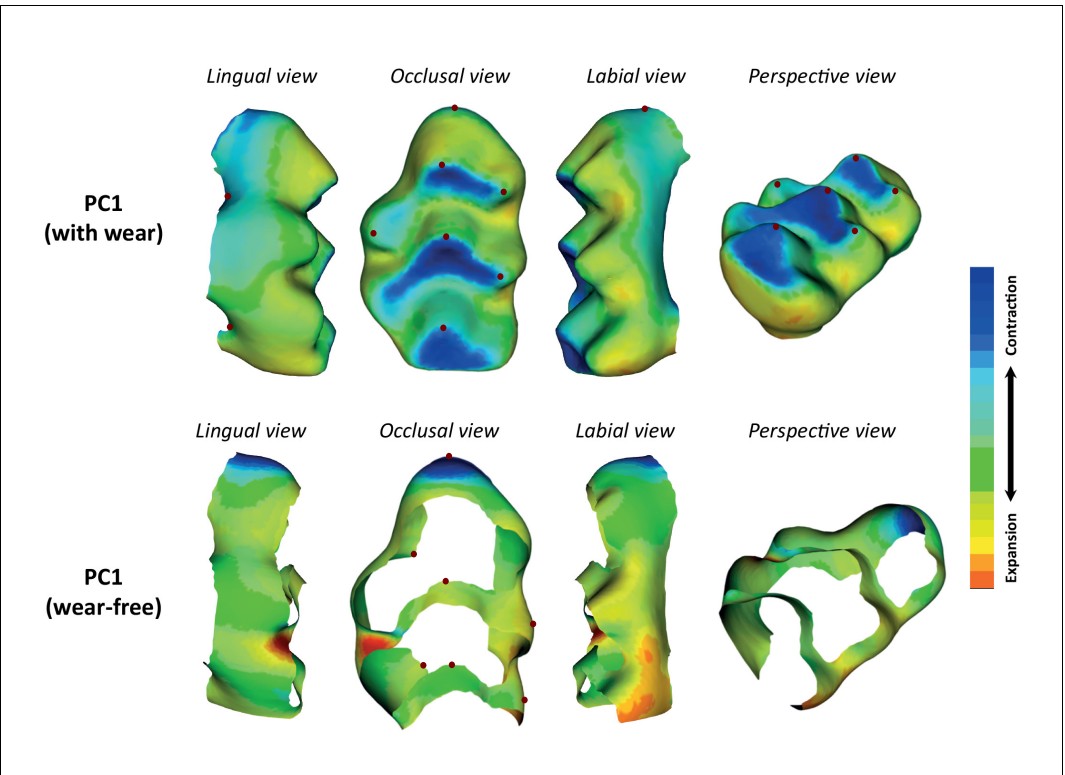

**Figure 2.** Effect of age and wear in molar shape variation. The molar shape of all hybrid mice was measured using the complete template, and the wear-free template. The shape reconstruction of the first principal component (PC1) derived from the complete template, and the wear-free template are shown. Abrasion of the cusps is evident in the complete template indicating wear effects. The landmarks used to anchor the template to the tooth surface are shown as red dots.

DOI: https://doi.org/10.7554/eLife.29510.004

Materials and methods) and contain, together, 15 protein coding genes. Mo.X.2 is 63.8 Mb given the strong LD pattern around the best SNP and contains 306 protein coding genes. Together, the loci explain ~10% of molar shape variation, with individual effects ranging from 1% to 3% (*Table 3*). The shape changes associated with each region were estimated as the difference between the consensus shape of the two homozygous states for each SNP. The phenotypic effect of each SNP is therefore not restricted to the PC it was associated with, but represents the effect of the SNP on the complete shape dimensionality (*Figure 4*-b).

### *Mitf* gene affects molar shape in mice

From the genes located in the QTL regions, two have been reported to affect tooth development directly or indirectly (MGI database queried 22.03.16), namely *Rps6ka3* (*Laugel-Haushalter et al., 2014*) and the microphthalmia-associated transcription factor *Mitf* (*Al-Douri and Johnson, 1987*). The *Rps6ka3* gene is located in Mo.X.2 that contains 305 additional genes, making it difficult to assess its relevance. However, *Mitf* is one of two genes found in Mo.6, and since many different mutations are known in this gene, it is possible to determine its relevance in tooth development.

*Mitf* is mainly known for its role in melanocyte development and proliferation in mice and humans. Mice carrying *Mitf* mutations show reduced or absent pigmentation, reduced eye size (microphthalmia), and deafness (*Steingrimsson et al., 2004*). Some mutations result in defective bone resorption due to defects in osteoclast development (*Hodgkinson et al., 1993*; *Moore, 1995*; *Hershey and Fisher, 2004*); the *Mitf^{mi}* allele leads to lethality at 3 weeks of age due to severe osteopetrosis. Abnormal morphology and eruption of incisor and molar teeth has been reported in some mutants (*Al-Douri and Johnson, 1987*; *Steingrimsson et al., 2002*), although this is thought to be linked to severe osteopetrosis in the mandible of these mutants.

**Table 2.** SNP heritability estimates per principal component axis.

The standard error of the estimate derived from LMM in GEMMA is shown. The heritability of molar shape is a weighted sum of the heritability per PC, the weights being the percentage of total variation represented by each PC.

| PC | %var | Heritability per PC | Error | Molar herit |
|----|------|---------------------|-------|-------------|
| 1 | 18.9 | 0.83 | 0.09 | 15.8 |
| 2 | 15.4 | 0.95 | 0.08 | 14.7 |
| 3 | 9.9 | 0.78 | 0.10 | 7.7 |
| 4 | 7.1 | 0.62 | 0.14 | 4.4 |
| 5 | 5.9 | 0.49 | 0.12 | 2.9 |
| 6 | 5.1 | 0.83 | 0.10 | 4.2 |
| 7 | 3.9 | 0.53 | 0.12 | 2.1 |
| 8 | 3.2 | 0.89 | 0.11 | 2.8 |
| 9 | 2.8 | 0.89 | 0.09 | 2.5 |
| 10 | 2.5 | 0.86 | 0.13 | 2.2 |
| 11 | 2.1 | 0.72 | 0.14 | 1.5 |
| 12 | 1.8 | 0.60 | 0.15 | 1.1 |
| 13 | 1.7 | 0.68 | 0.13 | 1.2 |
| 14 | 1.5 | 0.56 | 0.16 | 0.8 |
| 15 | 1.4 | 0.53 | 0.15 | 0.7 |
| 16 | 1.3 | 0.26 | 0.16 | 0.3 |
| 17 | 1.2 | 0.14 | 0.20 | 0.2 |
| 18 | 1 | 0.51 | 0.14 | 0.5 |
| Total Var | 86.7% | | | |
| | | pve for molar shape | | 65.50 |

DOI: https://doi.org/10.7554/eLife.29510.005

Interestingly, the large Mo.X.2 region, which is associated with the same PC as *Mitf*, includes *Ap1s2*. This gene is a target of the *Mitf*-regulated miRNA *miR-211*, and its function has been validated in the context of melanoma (**Margue et al., 2013**) and osteosclerosis in humans (**Saillour et al., 2007**).

To further evaluate a possible role of *Mitf* in molar shape variation, we examined molar morphology in mice carrying four mutant *Mitf* alleles. To avoid indirect effects on molar morphogenesis, we selected alleles for which no, or very mild osteopetrosis has been described (**Table 1**). $Mitf^{mi}$ induces severe osteopetrosis when homozygous (**Steingrimsson et al., 2002**). However, we have only used it in the heterozygous state. The other mutant alleles do not exhibit osteopetrosis in homozygotes (**Steingrimsson et al., 2002**; **Steingrimsson et al., 2004**).

As shown in **Figure 5**, all *Mitf* alleles affected the shape of the upper first molar. Pairwise tests between wild type B6 mice and each mutant group showed significant differences in mean shape (**Supplementary file 1A**). In terms of Procrustes distances, $Mitf^{mi-enu22(398)}$ is closest to wild type (0.0237), and $Mitf^{mi-vga9}/Mitfmi^{mi-vga9}$ is most distant (0.0466) (**Supplementary file 1A**).

$Mitf^{Mi-wh}/Mitf^{mi}$ compound heterozygotes have a similar molar shape as $Mitf^{Mi-wh}/+$ heterozygotes, suggesting that the $Mitf^{mi}$ allele alone has no additional effect on molar shape (**Figure 5d,g**). Interestingly, in homozygous condition, $Mitf^{mi}$ does result in severe pigmentation and eye phenotypes (**Steingrimsson et al., 2004**). However, given that $Mitf^{mi}/+$ mice were not examined in this study, we are not able to rule out the role of this allele in tooth shape. The molar shape generated by $Mitf^{mi-vga9}/+$ is very similar to $Mitf^{Mi-wh}/Mitf^{Mi-wh}$, although the intensity of the effect seems stronger in the latter (**Figure 5b,e**). For the $Mitf^{mi-vga9}$ and $Mitf^{Mi-wh}$ alleles, three genetic combinations were available. The $Mitf^{mi-vga9}$ mutation behaves additively regarding molar shape (**Figure 5**).

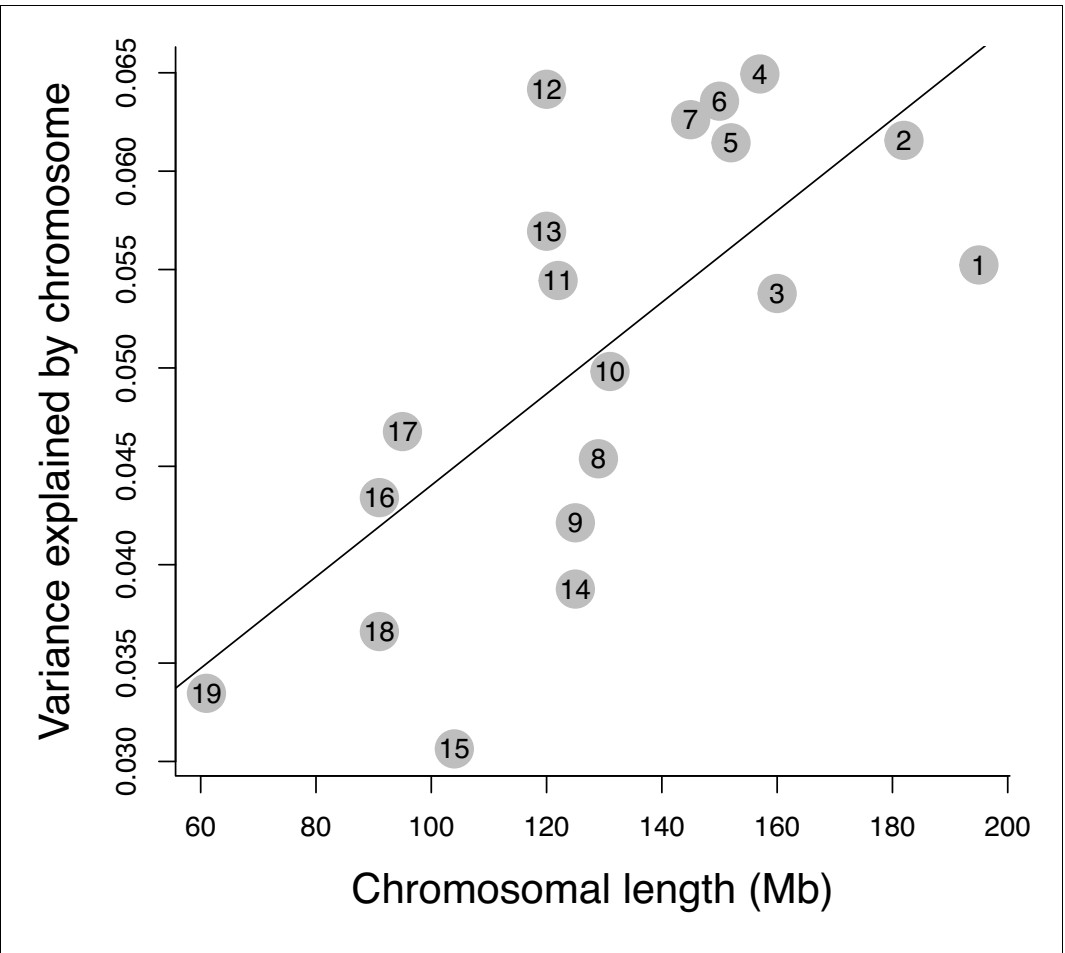

**Figure 3.** Relative effect of chromosomes on molar shape variance. The correlation between length and effect size of the 19 autosomes is shown (p=0.001, r = 0.67).

DOI: https://doi.org/10.7554/eLife.29510.006

This contrasts with reports on other phenotypes where its effect is clearly recessive (*Hodgkinson et al., 1993*).

A comparison between each mutant group and Mo.6, the region where *Mitf* gene is found, shows that mice homozygous for the $Mitf^{Mi-wh}$ allele are most similar to Mo.6 with respect to tooth shape. $Mitf^{mi-vga9}/Mitfmi^{mi-vga9}$ follows second in correlation strength (*Figure 5h*). The magnitude of shape variation generated by the mutant alleles falls within the range of shape variation observed in the mapping population (*Figure 5i*). However, it should be noted that the shape variation represented by the hybrid mice used here is determined by many genes, while the shape of mutant mice is the result of a mutation in a single gene.

## Discussion

Using genome-wide association mapping and 3D surface morphometrics we were able to map genomic regions underlying shape variation in the first upper molar of wild mice. The high mapping resolution achieved enabled identification of individual candidate genes making it feasible to functionally evaluate such candidates. Using a panel of mice with different mutant genotypes, we showed that one of these candidates, the transcription factor *Mitf*, has significant effects on molar shape.

The mapping population and the phenotyping approach used here played a definitive role in the ability to identify the genetic basis of molar shape variation. The mice were derived from wild-caught parents collected in the Bavarian hybrid zone in Germany (*Turner et al., 2012*) and therefore are

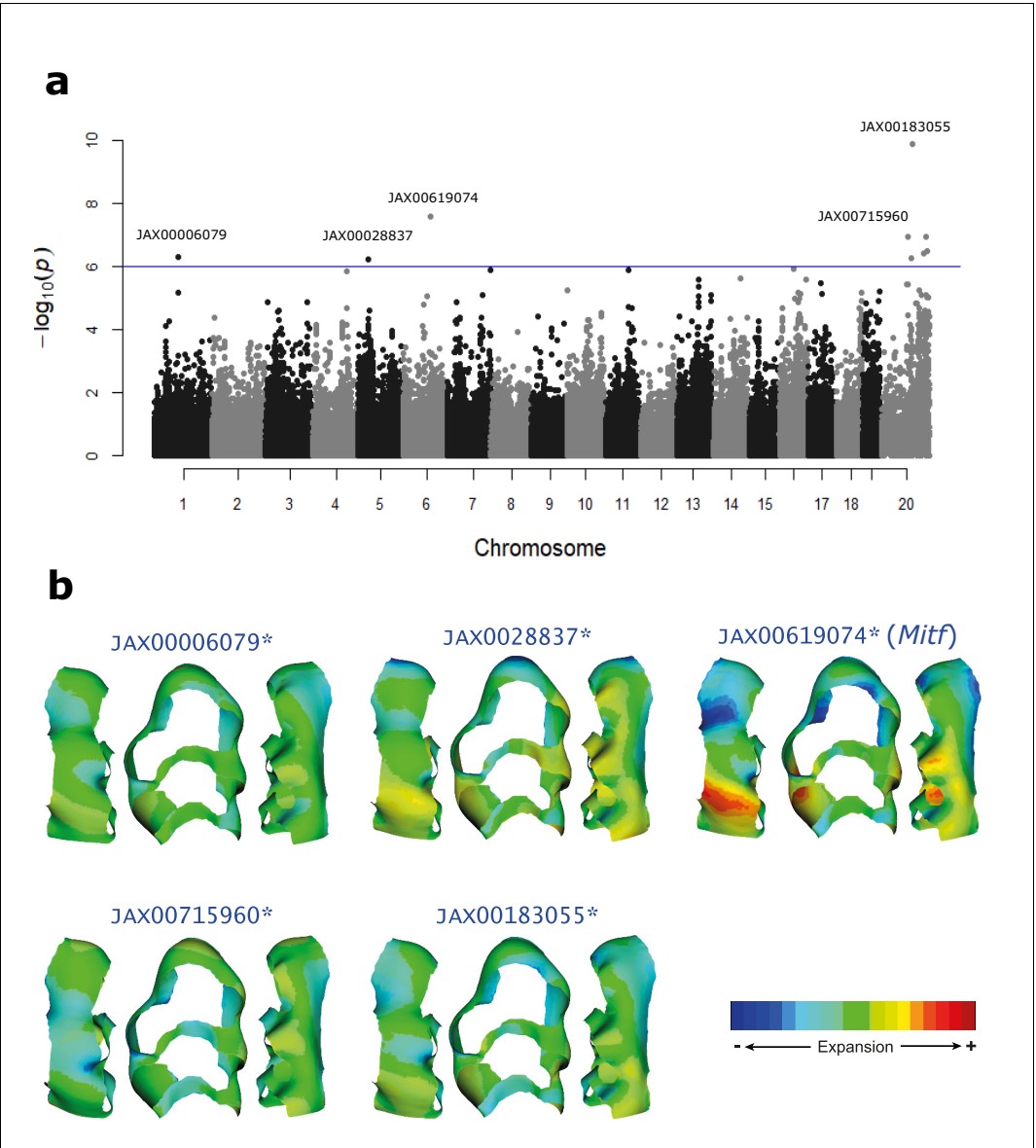

**Figure 4.** Genomic loci associated with molar shape variation. (**a**) Manhattan plot showing SNPs associated with molar shape variation. The blue line indicates the genome-wide significance threshold ($1 \times 10^{-6}$). However, to determine significance, a threshold was derived by permutations for each PC, and independently for autosomes and X chromosome (see Materials and methods). (**b**) Molar shape variation associated with the most significant SNP within each locus, estimated as the shape difference between the two homozygous SNP states. Warm colors indicate expansion and cold colors indicate compression of tissue relative to the mean shape. The SNP associated with the gene *Mitf* shows stronger localized effects. The raw phenotypic data used for the association mapping can be found as Source Data 1.

DOI: https://doi.org/10.7554/eLife.29510.007

The following source data is available for figure 4:

**Source data 1.** Shape data used in the association mapping.

DOI: https://doi.org/10.7554/eLife.29510.008

representative of wild genomic and phenotypic variation. The phenotypic changes driving the first axis of variation do not correspond to the shape changes between *M.m.musculus* and *M.m.domesticus*, indicating that this sample contains not only between-species patterns of variation but also strong within-subspecies variation. Such variation can be the result of transgressive segregation or

**Table 3.** Association mapping of molar shape variation.

The name of each significant region is defined by Mo (molar) and chromosomal location. The SNP with lowest p-value, its position in the genome, and p-value are shown. Effect size is calculated as the percentage of molar shape variation explained by the SNP. *All protein coding genes in the significant regions are shown, except for region Mo.X.2 where only genes relevant to the discussion are included; in total it contains 306 protein-coding genes. **Only this region was associated with more than one SNP. The five significantly associated SNPs spam a 55 Mb region (ChrX:104533418–15959832).

| QTL | Chr | Position | Best SNP | p-value | Effect size | PC axis | Genes* |
|-----|-----|----------|----------|---------|-------------|---------|--------|
| Mo.1 | chr1 | 84306638 | JAX00006079 | 5.11E-07 | 1.1% | PC11 | Pid1, Dner |
| Mo.5 | chr5 | 36723779 | JAX00128837 | 5.79E-07 | 3.2% | PC18 | Psapl1, Tada2b, Ccdc96, Grpel1, Tbc1d14, D5Ertd579e, Sorcs2 |
| Mo.6 | chr6 | 97980057 | JAX00619074 | 2.71E-08 | 2.8% | PC7 | Gm765, Mitf |
| Mo.X.1 | chrX | 92638616 | JAX00715960 | 1.18E-07 | 1.6% | PC16 | Fam123b, Zc4h2, Asb12, Arhgef9 |
| Mo.X.2** | chrX | 104533418 | JAX00183055 | 1.28E-10 | 2.2% | PC7 | Rps6ka3, Dach2, Ap1s2, Itm2a and 301 other genes |

DOI: https://doi.org/10.7554/eLife.29510.009

developmental instability, often associated with hybridization. However, the latter seem to play a very small role in this population (*Pallares et al., 2016*). This is the first time the genetics of molar shape variation in mice has been studied using wild mice, offering an evolutionary relevant perspective for the understanding of phenotypic variation. The technical advantage comes from the fact that these mice are hybrids between *M.m.musculus* and *M.m.domesticus*. These subspecies have been hybridizing in the Bavarian region for around 3000 years (reviewed in *Baird and Macholán, 2012*) resulting in high mapping resolution as a consequence of a genomic landscape where LD blocks are much smaller compared to traditional QTL-mapping approaches which usually correspond to two generations of crossing (*Workman et al., 2002*). The second technical advantage comes from the reduction of environmental variation; the mice were raised in the laboratory under controlled conditions and therefore the relative effect of genetic variation is enhanced at the expense of environmental variation. The suitability of this population for mapping loci associated with complex traits has been demonstrated previously for craniofacial shape (*Pallares et al., 2014*) and sterility phenotypes (*Turner and Harr, 2014*).

The phenotyping approach used here made use of 3D surface morphometrics, allowing us to quantify additional dimensions of variation compared to the two-dimensional approach (*Workman et al., 2002*; *Shimizu et al., 2004*). By measuring the surface of the tooth, this approach captures variation generated by differential wear between individuals, a confounding factor that is not present in 2D studies. Following *Ledevin et al. (2016)*, we used a wear-free template that allowed us to preserve the additional shape information captured by 3D methods and exclude wear-related variation. In this way, we were able to identify genetic loci associated with shape variation that otherwise would have been obscured by strong wear effects (see Results). Our findings are, however, a subset of the possible associations between molar shape and genomic regions. Between-individual differences in cusp shape remain to be explored at the genomic level.

## Genetic architecture

We have shown that more than half of the molar shape variation can be attributed to additive genetic effects, and that its genetic architecture is indeed polygenic, with many loci of small-to-moderate effect fine-tuning the phenotype within species. The same architecture has been reported for skull and mandible shape, traits that differ from teeth in their origin and time of development (*Pallares et al., 2014*; *Pallares et al., 2015*). The only study, up to now, addressing the genetic basis of molar shape variation in mice identified 18 QTL for 2D molar row shape using a F2 cross between LG/J and SM/J inbred mouse lines (*Workman et al., 2002*). This result points toward the polygenicity of molar shape determination. However, the focal trait (molar row) and low mapping resolution limit the comparisons with our findings. Using a modeling approach, *Salazar-Ciudad and Jernvall (2010)* have suggested that population-level variation in molar shape might have a simple genetic basis since they are able to recreate it by tuning a small number of parameters in the model. However, it cannot be excluded that such parameters are essentially polygenic, and therefore, even

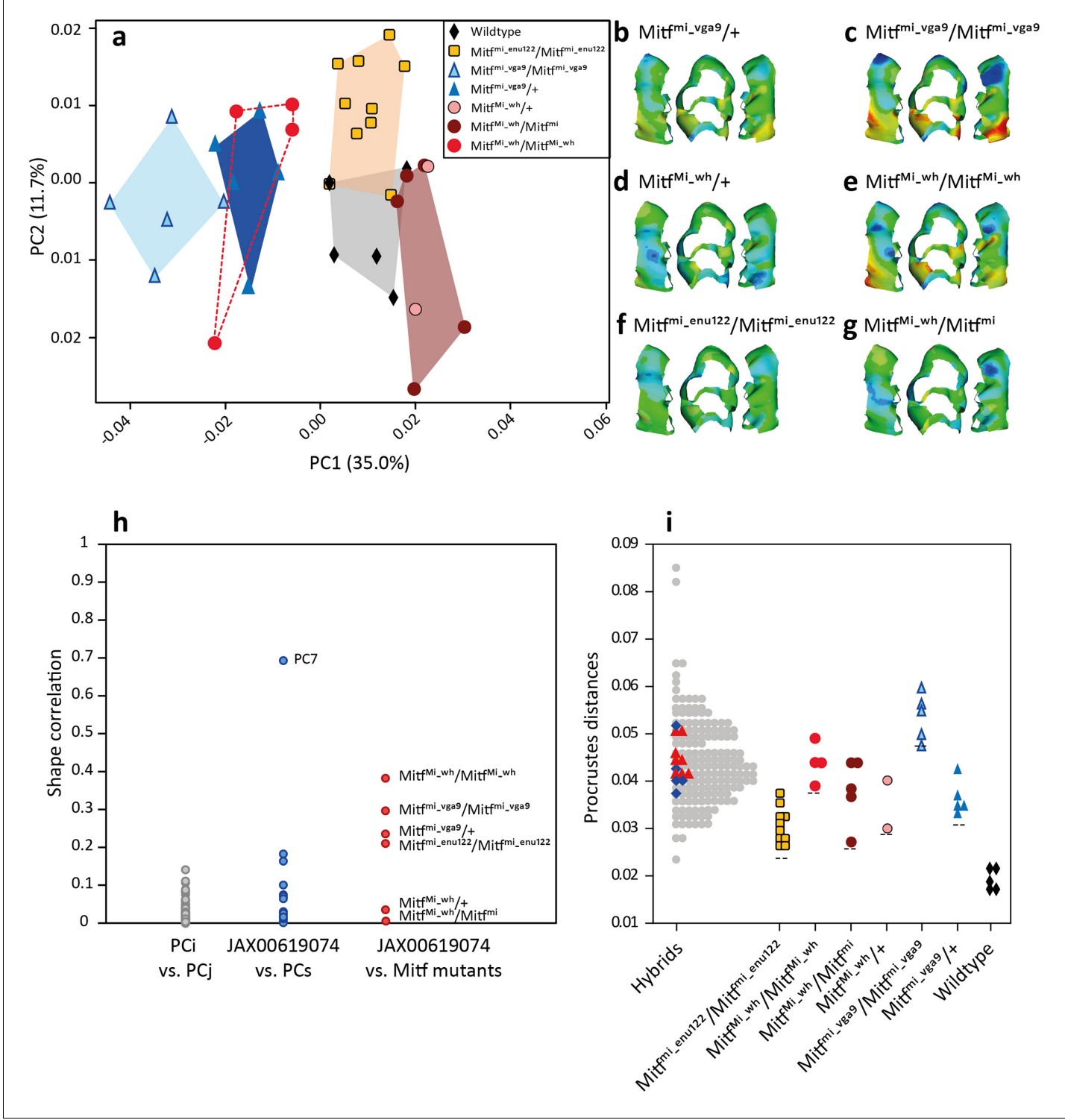

**Figure 5.** The effects of *Mitf* mutant alleles on the shape of the upper first molar. (a) Differentiation of the mutants from the wild-type (B6) in a morphospace (first two axes of a PCA on molar shape descriptors). (b–g) Mean phenotypic effect of each mutation relative to the mean shape of wild-type (C57Bl/J6) mice. The same color scale was used for all shape reconstructions. Warm colors indicate expansion and cold colors compression. (h) Correlation between various effects on tooth shape. Grey dots: pairwise correlations between PCs used for the mapping. They provide a proxy for the expected correlation between orthogonal directions of change. Blue dots: comparison between PCs used for the mapping and the effect of the SNP associated with *Mitf*-JAX00619074 SNP (most significant SNP in Mo.6 region). Only PC7, associated with Mo.6 in the mapping, resembles the shape effect of the *Mitf*-associated SNP. Red dots: comparison of the effect of *Mitf* alleles with the JAX00619074 SNP. Most *Mitf* mutants display an effect on

*Figure 5 continued on next page*

Figure 5 continued

tooth shape correlated with the effect of Mo.6. (i) Tooth shape variation in the hybrid mice used in the GWAS, and shape variation in the *Mitf* mutants. Each point corresponds to the Procrustes distance between a mouse tooth and a consensus shape. For hybrid mice the consensus shape is the mean tooth shape of all hybrids; for mutant mice it corresponds to the mean shape of wild-type mice. Within the hybrid group, blue diamonds and red triangles represent individuals with more than 80% alleles from *M.m.musculus* or *M.m.domesticus*, respectively. The *Mitf* mutations studied here generate shape changes in the range of magnitude (Procrustes distance) of natural variation observed within wild hybrids.
DOI: https://doi.org/10.7554/eLife.29510.010

when the model provides valuable insights into covariation of molar traits, it may not be adequate to decipher the genetic architecture of the trait (*Urdy et al., 2016*).

Despite the significant amount of genetic variation underlying phenotypic variation, we were only able to identify five genomic regions significantly associated with four (of 18) PCs. Together these loci explain around 10% of the total phenotypic variation, hence there are additional contributing loci yet to be identified. This is expected for a highly polygenic trait; increasing sample size and implementing multivariate mapping of shape traits will likely identify additional genes contributing to natural variation in molar shape.

## *Mitf* and molar shape variation

Mutations in several genes have been shown to affect cusp patterning in mice, most of them generating large changes in teeth morphology (reviewed in *Jernvall and Thesleff, 2012* and *Bei, 2009*). In contrast, to our knowledge, there are no reports of loci generating variation in molar shape of similar magnitude to variation observed in the wild, where shape differences between individuals are small. Here, we report that the transcription factor *Mitf*, a candidate gene identified by association mapping, affects the shape of the upper first molar in mice. All *Mitf* mutant alleles that were studied here had a significant effect on molar shape, regardless of severity of the mutation. Interestingly, even mice heterozygous for the mutant alleles showed consistent shape changes. The consistency in direction and magnitude of the phenotypic effect of each mutation (see *Figure 5*) suggests that the shape changes are indeed caused by the mutant allele, and are not the product of noise in the development of the tooth or associated tissues. In the latter case, we might have expected mice with the same mutant allele to exhibit non-consistent tooth shape changes. Although all mutations affect the phenotype, $Mitf^{Mi-wh}$ in homozygous state, most closely resembles the molar shape associated with Mo.6, the QTL where *Mitf* is found.

Severe osteopetrosis is associated with failures in tooth eruption (*Al-Douri and Johnson, 1987*; *Moore, 1995*), and this could suggest that the effect of *Mitf* on tooth shape is a byproduct of bone resorption deficiencies. However, as mentioned earlier, the alleles used in this study do not generate osteopetrosis (*Steingrimsson et al., 2002*; *Steingrimsson et al., 2004*), suggesting that the tooth phenotypes evidenced here are independent of the osteopetrotic effects of *Mitf*.

Some differences exist between the effects of *Mitf* on molar shape and previously reported phenotypes. For example, $Mitf^{mi-vga9}$ acts additively with respect to tooth shape; heterozygotes have a clear and distinct phenotype. This is different from the pigmentation phenotype and microphthalmia where the heterozygotes exhibit no visible phenotype (*Hodgkinson et al., 1993*; *Steingrimsson et al., 2003*). This indicates that the effect of some alleles is dependent on the organ or body region. However, it cannot be excluded that the way phenotypes are defined, for example, qualitatively vs quantitatively, is responsible for these discrepancies.

The evidence presented here for effect of *Mitf* on molar shape comes from mutations in a mouse laboratory strain, and it is therefore not equivalent to comparing the effect of naturally occurring alleles. This is evident from the large phenotypic effect of each *Mitf* mutant compared to within-population molar shape variation (*Figure 5i*). However, at least nine missense variants of *Mitf* segregate in wild mouse populations (*Harr et al., 2016*) (*Supplementary file 1B*). Whether such naturally occurring variants are associated with tooth shape variation remains to be tested. If this is indeed the case, the next step will be to explore how this gene is integrated into the already known pathways controlling tooth morphogenesis. Moreover, it will need to be assessed whether polymorphism in this gene is the result of neutral processes or the result of natural selection. Given the importance of *Mitf* in various pathways, e.g. pigmentation and ossification, the subtle molar shape variation generated at the intraspecific level might be a byproduct of a polymorphism maintained by

its role on other phenotypes, not directly on tooth shape. This type of pleiotropic effects on molar teeth has been proposed elsewhere regarding conspicuous morphological changes (*Rodrigues et al., 2013*).

## Concluding remarks

We have shown that subtle phenotypic variation at the micro-evolutionary level (i.e within-species) has a strong additive genetic basis. Such variation in molar shape is due to many small effect loci, but identification of loci and validation of causative genes is feasible. We report that the candidate gene, *Mitf*, has subtle but consistent effects on molar shape. We expect the results presented here to serve as a framework to further explore the way in which small effect loci act together to generate a functional, but still variable morphological shape.

# Acknowledgements

We are indebted to Diethard Tautz for many discussions during the development of this project and for funding support through institutional funds of the Max Planck Society. We thank Elke Blohm-Sievers for scanning the mouse samples. SR, SP, and RL were funded by the Agence Nationale de la Recherche, project Bigtooth (ANR-11-BSV7-008). ES was funded by a grant from the Icelandic Research Fund (#152715–053).

# Additional information

## Funding

| Funder | Grant reference number | Author |
| --- | --- | --- |
| Agence Nationale de la Recherche | Bigtooth (ANR-11-BSV7-008) | Ronan Ledevin Sophie Pantalacci Sabrina Renaud |
| Icelandic Research Fund | 152715-053 | Eirikur Steingrimsson |

The funders had no role in study design, data collection and interpretation, or the decision to submit the work for publication.

## Author contributions

Luisa F Pallares, Conceptualization, Formal analysis, Investigation, Writing—original draft, Writing—review and editing; Ronan Ledevin, Conceptualization, Formal analysis, Investigation, Writing—review and editing; Sophie Pantalacci, Conceptualization, Writing—review and editing; Leslie M Turner, Eirikur Steingrimsson, Conceptualization, Resources, Writing—review and editing; Sabrina Renaud, Conceptualization, Supervision, Writing—review and editing

## Author ORCIDs

Luisa F Pallares  http://orcid.org/0000-0001-6547-1901
Leslie M Turner  http://orcid.org/0000-0002-5105-3546
Eirikur Steingrimsson  http://orcid.org/0000-0001-5826-7486
Sabrina Renaud  http://orcid.org/0000-0002-8730-3113

## Ethics

Animal experimentation: The mice used for the association mapping were previously used in another study; details on animal experiment and ethics can be found in the original publication Turner et. al. 2012 Evolution. The mutant mice used for the validation of the gene Mitf were raised at the University of Iceland, BioMedical Center, under permit number 2013-03-01 from the Committee on Experimental Animals (Tilraunadýranefnd).

## Decision letter and Author response

Decision letter https://doi.org/10.7554/eLife.29510.015

Author response https://doi.org/10.7554/eLife.29510.016

## Additional files

### Supplementary files

• Supplementary file 1. (A) Shape comparison between *Mitf* mutants and wild-type B6 mice. A Hotelling $T^2$ test was performed to evaluate the difference in mean shape between mutant and wildtype groups; p-value, test statistic, and sample size (N) are shown. The Procrustes distances between mutant and wild type mean shapes are also indicated. *The comparison between heterozygous and homozygous mice for the $Mitf^{mi-vga9}$ mutation is also shown. (B) Missense variants found in the gene *Mitf* of wild mice. Nine populations of wild mice were screened for SNPs causing coding changes in Mitf: *Mus musculus musculus* from Kazakhstan, Check Republic, and Afganistan; *M. m. domesticus* from Iran, Heligoland, France, and Germany; *Mus castaneus;* and *Mus spretus.* Data available in the UCSC browser → MyData - > Public Sessions - > wildmouse (*Harr et al., 2016*). In addition, eight hybrids between *M.m. musculus* and *M.m. domesticus* from the German hybrid zone (Turner, Tautz and Harr unpublished data), the same population used in this study, were also screened for coding changes. Reference and variant alleles are shown. nVar = number of chromosomes with the variant allele, n = number of chromosomes per population.
DOI: https://doi.org/10.7554/eLife.29510.011

• Transparent reporting form
DOI: https://doi.org/10.7554/eLife.29510.012

### Major datasets

The following previously published dataset was used:

| Author(s) | Year | Dataset title | Dataset URL | Database, license, and accessibility information |
|---|---|---|---|---|
| Pallares LF, Turner LM, Harr B, Tautz D | 2014 | Use of a natural hybrid zone for genome-wide association mapping of craniofacial traits in the house mouse | http://dx.doi.org/10.5061/dryad.bt848 | Available at Dryad Digital Repository under a CC0 Public Domain Dedication |

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
