## [Decision Letter]

Thank you for submitting your article "Genomic regions controlling shape variation in the first upper molar of the house mouse" for consideration by *eLife*. Your article has been favorably evaluated by Patricia Wittkopp (Senior Editor) and three reviewers, one of whom, Craig T Miller (Reviewer #1), is a member of our Board of Reviewing Editors. The following individual involved in review of your submission has agreed to reveal their identity: Alistair R. Evans (Reviewer #3).

The reviewers have discussed the reviews with one another and the Reviewing Editor has drafted this decision to help you prepare a revised submission.

Summary:

Pallares and colleagues study mouse molar shape variation in two subspecies of house mice. Using a GWAS and 3D surface morphometric approach in hybrid mice, the authors identify five genomic regions associated with molar shape. One genomic region contains the candidate gene *Mitf*, and the authors show in lab mice with different mutant alleles of *Mitf* a role for *Mitf* in regulating molar shape. Overall this study presents the first GWAS for molar shape in natural populations, identifies the first genomic regions found to underlie quantitative variation in molar shape in nature, and identifies a specific gene (*Mitf*) regulating quantitative aspects of molar shape in the lab that appears to also regulate natural variation in tooth shape.

Essential revisions:

1) One potential concern is many of the *Mitf* mutant comparisons are done between mutant mice and a panel of non-sibling B6 mice. It seems possible that differences in genetic background in these mouse lines contribute to the differences in tooth shape. One nice argument against this is the authors observe differences in tooth shape between mice heterozygous and homozygous for the mi-vga9 allele. It would strengthen the authors' conclusions if in Table 1 they compared the mi-vga9/mi-vga9 mice not just to wild-type mice, but to the mi-vga9/+ mice. In Figure 5, these genotypes look quite different, but it would be better to also formally test the differences. Along these lines, can the authors comment on whether the mi-vga9/mi-vga9 and mi-vga9/+ mice were full siblings? If so, that would further strengthen the argument that *Mitf*, not genetic background differences are responsible.

2) Although finding actual mutations underlying the *Mitf*-associated QTL is beyond the scope of this study, can the authors at least comment on whether coding changes are found between the *domesticu*s and *musculus* mice used in their study, or whether *Mitf* coding changes are present in the reference genome assemblies available for *domesticus* and *musculus* derived strains or populations?

3) Subsection “Association mapping”, third paragraph: Can the authors further justify the decision to pick arbitrary 500 kb intervals for most QTL? It seems they have data that would speak to patterns of LD in this sample, including genome-wide averages and within these QTL, so not sure whether 500 kb is an overly conservative estimate or not.

4) Subsection “Association mapping”, second paragraph: I was surprised that no genomic regions were significantly associated with size variation, given how polygenic and strong the signal appears for tooth shape. Was mouse size corrected for in mapping centroid size?

5) "The other mutant alleles do not exhibit osteopetrosis in homozygotes." Can the authors clarify for this statement, whether they mean osteopetrosis has not been reported in homozygotes for these alleles, or whether they (or other groups) looked for osteopetrosis in homozygotes? If the latter, the authors should provide references or data to back up this claim. If the former, the sentence should be edited. Same comment for “The evidence presented here for effect of *Mitf* on molar shape comes from mutations in a mouse laboratory strain, and it is therefore not equivalent to comparing the effect of naturally occurring alleles”.

6) Materials and methods: Only the first upper molar is analyzed. Why not, but the rational for this choice is never explained through the manuscript.

7) Geometric morphometrics: You used 10 landmarks to anchor the template. They are very important features to know because they impose some constrains, but there is no way for the readers to get this information.

For the full template, 1588 semi-landmarks are used and for the "wear-free" 1532. This is much denser in the second case than in the first given that all tips of cusps are removed. I don't understand why you increase the sampling.

8) Subsection “Tooth shape, mouse age and wear”. Your idea is that age is related to wear, as once erupted tooth shape doesn't change except by the effect of wear. In my opinion you should explain it to readers.

9) Wear is expected to be on PC1. People have used Burnaby-like procedure to get rid of such artifacts (many examples with fish) based on PC1 or expected shape changes. You decide to use a very different alternative approach (quite a disappointing one) by cutting cusp tips. Why not, but why? Because age impacts a lot of PCs and thus you consider that the effect of wear is spread all over the shape space? If so, I think you should explain your rational. Interestingly, age explain the same amount of variance with the 'wear-free' template but is less significant.

10) Finally you analyzed something very close to what is captured with 2D outline. The amount of 3D information you have in your sampling depend on the cusp height kept in the template. You didn't say anything about that. How did you decide the height at which you cut the tips?

11) I don't understand why you run separate Procrustes superimpositions (subsection “Functional evaluation of the candidate gene *Mitf*”, second paragraph). It adds a very complex way of comparing shape changes. I can understand that you were afraid that weird mutant twists your Procrustean space for GWAS but in my opinion, you could have added the mutant as supplementary observations into the hybrid GPA (i.e. superimposing each mutant to the hybrid mean shape). It will have allowed the comparison of shape changes using more classical tools of geometric morphometrics (angles between shape vectors, vector correlation etc.). In your approach, I don't get how the two can be centered and aligned to ensure that trivial variation are removed.

12) One additional remark on this method of comparing shape changes is that in order to get a threshold you consider the values obtained between PCs. PCs are orthogonal but not independent (subsection “Functional evaluation of the candidate gene *Mitf*”, last paragraph). The math ensures that they are orthogonal but nothing ensure that they are independent given the underlying generative processes (for ex genetics).

13) GWAS: The family structure in your sample is very strong and you correctly use linear mixed model to handle it but, as in your previous papers, running multiple univariate LMM on PCs is not the most powerful approach, but I understand that running multivariate LMM is challenging, but the cost is evident here as you don't have a very powerful design with 183 mice.

"To facilitate association mapping" but it is at the price of some power lost because PCs are not aligned to the underlying genetics.

Nonetheless, you correctly want to assess the effect size of best SNPs simultaneously on the full shape space (on all tangent coordinates), but why using only the first 18 PCs? Why estimating these effect sizes only on 86% variance? Did you consider that 14% are just analytical errors?

14) About that (subsection “Association mapping”, last paragraph) you say that you used the coefficient of determination but with multivariate data I guess you used an analogue based on distances (a ratio of sums of sum-of-squares) but there are other analogue based on classical multivariate statistics (Pillai etc.). It may be better to precise that it is procrustes distances based R2.

15) It is unclear that you used leave-one chromosome out approach to handle the pop structure in your LMM (subsection “Association mapping”, second paragraph).

16) SNP heritability: subsection “SNP heritability”, first paragraph: In my opinion, the way you present your computation of the "total heritability" is right in calculus but is little weird in term of genetics. In my opinion, you assess the sum of the additive genetic variances and do the ratio with the total variance to get a quite poor proxy of h2, and indeed means not much in term of heritability (see discussions of Monteiro and Klingenberg between 2002 and 2010).

17) Functional evaluation: subsection “Samples used to functionally evaluate *Mitf*”: I don't agree with this statement about sexual dimorphism. There are plenty of phenotypes with SD in mammals and more specifically in the house mouse (see for example Kart et al. 2016 Nat Comms). The fact that SD is low on bone shape doesn't insure that it is the case in tooth.

18) Results, first paragraph: The most parsimonious hypothesis is simply it is within species variation and that the inter-subspecific variance is smaller than the within, species-specific loci have small effects on *this* character and are not distinguishable from within-species loci. I don't think we need to ask for transgressive variation or hybrid instability.

19) Discussion: subsection “Genetic architecture”, first paragraph: Actually this conclusion is based on the huge amount of missing heritability (difference between the SNP heritability and the actual loci you catch-up). However, as your GWAS doesn't properly handle the multivariate nature of the shape space, you don't know anything about that because for instance a locus, orthogonal to all PCs, and explaining 1% of variance on each, will finally explained a lot of variance but will never be captured with your approach.

20) You have a very small sample size to run Hotelling T^2^. Will you do a t-test with N = 5 samples? Same conditions apply to Hotelling. Moreover you have twice more additional parameters plus correlation between the two variables to estimate. Maybe doing something non-parametric based on distances will be more reliable.

21) Some p-values for statistical tests were not reported in the manuscript.

---

## [Author Response]

Essential revisions:1) One potential concern is many of the Mitf mutant comparisons are done between mutant mice and a panel of non-sibling B6 mice. It seems possible that differences in genetic background in these mouse lines contribute to the differences in tooth shape. One nice argument against this is the authors observe differences in tooth shape between mice heterozygous and homozygous for the mi-vga9 allele. It would strengthen the authors' conclusions if in Table 1 they compared the mi-vga9/mi-vga9 mice not just to wild-type mice, but to the mi-vga9/+ mice. In Figure 5, these genotypes look quite different, but it would be better to also formally test the differences. Along these lines, can the authors comment on whether the mi-vga9/mi-vga9 and mi-vga9/+ mice were full siblings? If so, that would further strengthen the argument that Mitf, not genetic background differences are responsible.

All the *Mitf* mutations are on a B6 background (this is stated in the Materials and methods section “Samples used to functionally evaluate Mitf”), and the B6 line used to generate the *Mitf* mutations is the same one used as control; this line is maintained in the same facilities and under the same conditions than the *Mitf* mutant lines derived from it.

The mi-vga9/mi-vga9 and mi-vga9/+ were indeed full siblings. This information was added in the first paragraph of the subsection “Samples used to functionally evaluate *Mitf*”. We have also added the comparison between mi-vga9/mi-vga9 and mi-vga9/+ in Table 1, demonstrating that differences between homozygous and heterozygous siblings was significant (p = 0.0076).

2) Although finding actual mutations underlying the Mitf-associated QTL is beyond the scope of this study, can the authors at least comment on whether coding changes are found between the domesticus and musculus mice used in their study, or whether Mitf coding changes are present in the reference genome assemblies available for domesticus and musculus derived strains or populations?

We compared *Mitf* sequences in samples from 6 wild populations of mice, including *M. m. musculus* and *M. m. domesticus, M. m. castaneus* and related *M. spretus* (Harr et al. Scientific Data 3, Article number: 160075 (2016) doi:10.1038/sdata.2016.75), and eight hybrid mice from the same population used in this study (Turner Tautz & Harr, unpublished data).

We found seven predicted missense variants in wild populations of mice. Five are found in *Mus spretus* and *Mus castaneus*. One variant is present in a population of *M.m. musculus* from Kazakhstan, and one in a population of *M.m. domesticus* in Germany. Another variant is present in mice from the hybrid zone used in this study.

We have added this information in the Discussion (last paragraph), and added a table of the predicted variants as Supplementary file 1.

The SNP data can be accessed in the UCSC genome browser (genome.ucsc.edu) -> My Data -> Public Sessions -> wildmouse

3) Subsection “Association mapping”, third paragraph: Can the authors further justify the decision to pick arbitrary 500 kb intervals for most QTL? It seems they have data that would speak to patterns of LD in this sample, including genome-wide averages and within these QTL, so not sure whether 500 kb is an overly conservative estimate or not.

In a previous study where we used the same SNP data (Pallares et al. 2014) we showed that the median size of the associated regions with LD > 0.8 was 150kb, and the regions with LD > 0.2 (weakly linked) was ~ 1.8 Mb. In the present study, some significant SNPs fell in regions with poor SNP density resulting in zero neighboring SNPs with LD> 0.8. Therefore, we decided to use the intermediate value 500 Kb here, similar to what we observed in the previous study, but not as stringent. We added a clarification of this choice in the third paragraph of the subsection “Association mapping”.

4) Subsection “Association mapping”, second paragraph: I was surprised that no genomic regions were significantly associated with size variation, given how polygenic and strong the signal appears for tooth shape. Was mouse size corrected for in mapping centroid size?

No, we didn’t account for the effect of body size on tooth centroid size. The first upper molar is mineralized around birth in mice, and although it might be correlated with body size at birth (we don’t have this data), it is not correlated with adult body size at the population level (Renaud et al. 2017, Mamm Biol). Using body size as a covariate might introduce variation that is actually not relevant for molar size. Besides, the lack of associations with size is not rare, usually such loci are of very small effect, and the studies that have been able to identify them are usually performed with F2 crosses of mouse inbred lines that were previously selected for body size differences.

5) "The other mutant alleles do not exhibit osteopetrosis in homozygotes." Can the authors clarify for this statement, whether they mean osteopetrosis has not been reported in homozygotes for these alleles, or whether they (or other groups) looked for osteopetrosis in homozygotes? If the latter, the authors should provide references or data to back up this claim. If the former, the sentence should be edited. Same comment for “The evidence presented here for effect of Mitf on molar shape comes from mutations in a mouse laboratory strain, and it is therefore not equivalent to comparing the effect of naturally occurring alleles”.

Osteopetrosis has been characterized for many of the *Mitf* mutations, the references for individual studies are found in the following review that we have now added as reference in the pertinent sections (subsection “*Mitf* gene affects molar shape in mice”, second, fourth and sixth paragraphs and subsection “*Mitf* and molar shape variation”, second paragraph): Steingrimsson et al., 2004, annual review genetics. And, we have also added, in the same sections, the following reference where the alleles mi, ew, Wh, and vga9 used in this study were directly tested for osteopetrosis: Steingrimsson et al. 2002, PNAS.

6) Materials and methods: Only the first upper molar is analyzed. Why not, but the rational for this choice is never explained through the manuscript.

We decided to focus on the first molar because it is the first one to develop in the molar row and therefore, it is relatively independent from the development of the second and third molars given that they develop following the Inhibitory-Cascade model (Kavanagh et al. Nature 2007). The second and third molar, in contrast, are not only a simplified version (in terms of morphological complexity) of the first molar, but are also directly affected/constrained by the development of the first one. Along these lines, the effect of genetic variation relative to environmental variation (in this case the development of other molars) on molar shape is enhanced on the first molar.

Regarding the lower molars, these are less morphologically complex compared to the upper molars. And therefore, exploring the first upper molar maximizes the amount of phenotypic variation available for mapping.

Given the results found here for the first upper molar we think it will be very interesting to explore the genetic basis of molar shape in the second and third molars independently of each other, but also in terms of the correlations that are created between them given the Inhibition-Cascade model. This however, will need to be explored in a different manuscript.

We added a line in the Introduction specifying that we focused on the first upper molar (fourth paragraph) and expanded on the reason for this in the second paragraph of the Materials and methods subsection “3D surface morphometrics”

7) Geometric morphometrics: You used 10 landmarks to anchor the template. They are very important features to know because they impose some constrains, but there is no way for the readers to get this information.For the full template, 1588 semi-landmarks are used and for the "wear-free" 1532. This is much denser in the second case than in the first given that all tips of cusps are removed. I don't understand why you increase the sampling.

The 10 landmarks were only used to pre-position the template to the original template before this was deformed to match the tooth surface. These landmarks were not used in any analysis and therefore they should not affect any of the morphometric results and interpretations. We have, however, added the landmarks to the surfaces on Figure 2, so it is clear for the reader where are they located.

The semi-landmarks are automatically sampled following a Poisson distribution with the aim of obtaining equally-spaced semi-landmarks. The number requested was 1500 for both templates with the aim of obtaining a dense sampling and therefore capture variation at a very small scale. To fulfill the equally-spaced requirement, the actual number selected by the algorithm is approximately the requested number, therefore the specific numbers used in each template are not identical. Since the number of semi-landmarks is very high, going from, e.g. 1500 in the full template to e.g. 1000 in the wear-free template won’t affect the observed results, and therefore we opted for using the same number for both.

We have now added in the Materials and methods, a statement regarding the number of requested landmarks requested and the number finally sampled (subsection “3D surface morphometrics”, third and fifth paragraphs).

8) Subsection “Tooth shape, mouse age and wear”. Your idea is that age is related to wear, as once erupted tooth shape doesn't change except by the effect of wear. In my opinion you should explain it to readers.

We have expanded the Results section “Tooth shape, mouse age and wear”to include the explanation for this. In short, yes, tooth shape doesn’t change once the teeth have erupted. Only wear affects tooth morphology after eruption, and this is why wear and age are correlated. However, other non-age-related factors like diet and behavior also affect the degree of wear, and this is one of the reasons why we used a wear-free template approach, to be able to address not only age-related wear patterns, but also other unknown factors.

9) Wear is expected to be on PC1. People have used Burnaby-like procedure to get rid of such artifacts (many examples with fish) based on PC1 or expected shape changes. You decide to use a very different alternative approach (quite a disappointing one) by cutting cusp tips. Why not, but why? Because age impacts a lot of PCs and thus you consider that the effect of wear is spread all over the shape space? If so, I think you should explain your rational. Interestingly, age explain the same amount of variance with the 'wear-free' template but is less significant.

Wear is a very difficult aspect to address especially because, as correctly suggested by the reviewers, it impacts several PCs. Age, although correlated with wear is an incomplete proxy for wear since there are other factors affecting wear patterns as described in the response to the previous comment and explored in Renaud and Ledevin 2017. Before choosing the wear-free method, we tried several approaches such as multivariate regressions, or using the dentine-enamel interface as the surface of interest. At the end, removing the cusp tips appeared to be the most convincing protocol. We also considered the fact that it should be possible to apply this phenotyping method to wild animals for which age is unknown (e.g. Ledevin et al., 2016).

10) Finally you analyzed something very close to what is captured with 2D outline. The amount of 3D information you have in your sampling depend on the cusp height kept in the template. You didn't say anything about that. How did you decide the height at which you cut the tips?

It is true that traditional 2D and our 3D phenotyping seem similar as they both focus on the crown, however, our 3D approach detects shape details that a 2D outline will not be able to detect. It captures information from the crown outline but also from the central parts of the tooth such as the inter-cusp space and the morphology of the cusp basal area. The slope at the front of the cusps, described by our 3D method, is not accessible with a 2D outline.

The height at which we cut the tip was decided empirically by comparing teeth of different wear stages. The cut section was designed so that the most worn teeth of our dataset can still be described by our template. We have now added this in the Materials and methods section (subsection “3D surface morphometrics”, fifth paragraph).

11) I don't understand why you run separate Procrustes superimpositions (subsection “Functional evaluation of the candidate gene Mitf”, second paragraph). It adds a very complex way of comparing shape changes. I can understand that you were afraid that weird mutant twists your Procrustean space for GWAS but in my opinion, you could have added the mutant as supplementary observations into the hybrid GPA (i.e. superimposing each mutant to the hybrid mean shape). It will have allowed the comparison of shape changes using more classical tools of geometric morphometrics (angles between shape vectors, vector correlation etc.). In your approach, I don't get how the two can be centered and aligned to ensure that trivial variation are removed.

Adding mutants as supplementary specimens is an option that we considered, however the probability for mutants would have been outside of the range of variation of the wild-derived mice. The risk of incorrect projection of the mutants into the shape space was therefore not negligible, and we preferred to run two separate analyses. Now, since there is not a Procrustes superimposition, there is not centering and alignment, and therefore our method doesn’t compare effects in the shape space, but directly on the tooth surface. The question we tried to answer in this way is whether two effects (e.g. mutant alleles, SNPs from the mapping) modify the tooth in the same way.

12) One additional remark on this method of comparing shape changes is that in order to get a threshold you consider the values obtained between PCs. PCs are orthogonal but not independent (subsection “Functional evaluation of the candidate gene Mitf”, last paragraph). The math ensures that they are orthogonal but nothing ensure that they are independent given the underlying generative processes (for ex genetics).

We have now re-written this part to make clear that the phenotypic changes associated with PCs are not necessarily independent given the underlying genetics. However, since it is not feasible to get totally independent shape changes to be used as a null in Figure 5, we still use the correlation between PCs as a reference for the correlation between orthogonal shape changes. We have however, been very clear regarding that a genetic correlation might exist between PC, and therefore we are not using such correlations as a “significance threshold” as was stated previously in the text, but just as a point of comparison. (subsection “Functional evaluation of the candidate gene *Mitf*”, last paragraph and legend Figure 5).

13) GWAS: The family structure in your sample is very strong and you correctly use linear mixed model to handle it but, as in your previous papers, running multiple univariate LMM on PCs is not the most powerful approach, but I understand that running multivariate LMM is challenging, but the cost is evident here as you don't have a very powerful design with 183 mice."To facilitate association mapping" but it is at the price of some power lost because PCs are not aligned to the underlying genetics.Nonetheless, you correctly want to assess the effect size of best SNPs simultaneously on the full shape space (on all tangent coordinates), but why using only the first 18 PCs? Why estimating these effect sizes only on 86% variance? Did you consider that 14% are just analytical errors?

We agree with this assessment. The mapping of univariate PCs is limited, and it is not the optimal way of mapping shape data. Whenever possible, shape data should be mapped in a multivariate manner. However, this was the way to circumvent the strong family structure of the data and therefore to be able to implement the LMM. We have now mentioned the limitations of this approach (subsection “Association mapping”, first paragraph) and removed the line “to facilitate association mapping”.

The choice of the first 18 PCs was totally arbitrary, as it usually is when it comes to decide on the “relevant” PCs. Here we just chose as a cutoff, the PCs that explained at least 1% of the variance.

14) About that (subsection “Association mapping”, last paragraph) you say that you used the coefficient of determination but with multivariate data I guess you used an analogue based on distances (a ratio of sums of sum-of-squares) but there are other analogue based on classical multivariate statistics (Pillai etc.). It may be better to precise that it is procrustes distances based R2.

Yes, we calculated r2 as the explained variance from sums of squares summed over all responses. We have clarified this in the third paragraph of the subsection “Association mapping”.

15) It is unclear that you used leave-one chromosome out approach to handle the pop structure in your LMM (subsection “Association mapping”, second paragraph).

We didn’t use the leave-one-chromosome-out approach. We used a unique kinship matrix including all chromosomes to evaluate associations in each chromosome. We however acknowledge that excluding the focal chromosome might have increased our power a bit.

16) SNP heritability: subsection “SNP heritability”, first paragraph: In my opinion, the way you present your computation of the "total heritability" is right in calculus but is little weird in term of genetics. In my opinion, you assess the sum of the additive genetic variances and do the ratio with the total variance to get a quite poor proxy of h2, and indeed means not much in term of heritability (see discussions of Monteiro and Klingenberg between 2002 and 2010).

The specific way in which heritability was estimated in this study is again, as a comment above adequately pointed out, a limitation associated with performing univariate mapping with a multivariate trait, and we acknowledge that. However, we don’t completely agree with the idea that estimating heritability for each shape dimension is the right approach for the question we want to answer in this study (referring to the Monteiro and Klingenberg discussion). We think that depending on the objective of the study (as we think it was the conclusion of Monteiro and Klingenberg discussion), a univariate estimate of variance is adequate. If the objective is, for example, to address response to selection, a scalar heritability value is of little use. But, if, as in this study, the intention is to get an impression on how much additive genetic variance underlies a trait, a scalar value is the only way of making it comparable with other studies or traits.

We have now clarified the intention behind calculating a scalar value for heritability (subsection “SNP heritability”, first paragraph).

17) Functional evaluation: subsection “Samples used to functionally evaluate Mitf”: I don't agree with this statement about sexual dimorphism. There are plenty of phenotypes with SD in mammals and more specifically in the house mouse (see for example Kart et al. 2016 Nat Comms). The fact that SD is low on bone shape doesn't insure that it is the case in tooth.

We totally agree with this comment, and with the results shown in the paper suggested by the reviewers. By no means we wanted to imply that sexual dimorphism, in general, is not important in mice. Thanks to the reviewer comment, we noticed that we had stated “sex has very small effect in *bone shape”*, however we meant *tooth shape*, and not bone shape. There’s evidence (we have cited such references, e.g. Valenzuela-Lamas et al. 2011, Renaud et al. 2017) that sexual dimorphism is very small, and most of the time is absent in molar tooth shape. Therefore, although our sample size is small and doesn’t allow us to test for sexual dimorphism, we are still confident that such factor doesn’t affect our results. We have now corrected the text (subsection “Samples used to functionally evaluate *Mitf*”, last paragraph).

18) Results, first paragraph: The most parsimonious hypothesis is simply it is within species variation and that the inter-subspecific variance is smaller than the within, species-specific loci have small effects on this character and are not distinguishable from within-species loci. I don't think we need to ask for transgressive variation or hybrid instability.

We agree with this interpretation. We have now added it to the text (Results, first paragraph), including reference where such pattern was described for bone shape. We have, however, kept transgressive phenotypes and hybrid instability as other sources of within-pop variation given the hybrid nature of the sample used in this study.

19) Discussion: subsection “Genetic architecture”, first paragraph: Actually this conclusion is based on the huge amount of missing heritability (difference between the SNP heritability and the actual loci you catch-up). However, as your GWAS doesn't properly handle the multivariate nature of the shape space, you don't know anything about that because for instance a locus, orthogonal to all PCs, and explaining 1% of variance on each, will finally explained a lot of variance but will never be captured with your approach.

Our conclusion has two lines of evidence. First, as the reviewers mention, the large amount of missing heritability. And second, the results derived from the chromosomal partitioning of the variance shown in Figure 3.

We agree with the fact that our univariate mapping approach misses genetic associations that are spread across many PCs, and this will affect the missing heritability estimates. However, the percentage of variance explained by a locus increases, but not in a dramatic way, when it is calculated based on the PC it is associated with, vs. on the combined effect of all PCs. We therefore, don’t expect single loci to explain a lot of variance in this phenotype, even if multivariate approach were used.

Yet, we offer another line of evidence for the polygenicity of tooth shape that is totally independent of the associations found in GWAS: the results of the chromosomal partitioning of the variance. This test has been used, mostly in human studies, to determine “how polygenic a polygenic trait is”. In this test, no individual associations between SNPs and phenotype are calculated, but the joint effect of all SNPs found in the different chromosomes. The rationale behind this is: if the effect of individual SNPs is small and many SNPs underlie phenotypic variation, there will be a linear correlation between chromosomal length and phenotypic effect. This is exactly the pattern that we see for tooth shape, and therefore, although we acknowledge and share the reviewers concerns regarding the first line of evidence, we are confident on the interpretation of our results.

We have added lines in the Results section to make clear the importance of the chromosomal-partitioning-of-the-variance analysis.

20) You have a very small sample size to run Hotelling T^2^. Will you do a t-test with N = 5 samples? Same conditions apply to Hotelling. Moreover you have twice more additional parameters plus correlation between the two variables to estimate. Maybe doing something non-parametric based on distances will be more reliable.

It is always tricky to test for differences when the sample size is small. However, given the circumstances (it was not possible to obtain more mutant mice), a t-test (here a T^2^) is one of the most performant tests given a small sample size (e.g. de Winter, PARE, 2013). More important than the significance of this test, is the distribution of the different mutant shapes that show a clear effect of each of the mutant alleles on molar tooth shape.

21) Some p-values for statistical tests were not reported in the manuscript.

We have now added the missing p-values in the last paragraph of the subsection “Tooth shape, mouse age and wear”, and subsection “Heritability”.